# The Emerging Role of Non-Coding RNAs in Pituitary Gland Tumors and Meningioma

**DOI:** 10.3390/cancers13235987

**Published:** 2021-11-28

**Authors:** Soudeh Ghafouri-Fard, Atefe Abak, Bashdar Mahmud Hussen, Mohammad Taheri, Guive Sharifi

**Affiliations:** 1Department of Medical Genetics, Shahid Beheshti University of Medical Sciences, Tehran 19835-35511, Iran; s.ghafourifard@sbmu.ac.ir; 2Men’s Health and Reproductive Health Research Center, Shahid Beheshti University of Medical Sciences, Tehran 19835-35511, Iran; atefeh.abak@gmail.com; 3Department of Pharmacognosy, College of Pharmacy, Hawler Medical University, Erbil 44001, Iraq; Bashdar.Hussen@hmu.edu.krd; 4Institute of Human Genetics, Jena University Hospital, 07743 Jena, Germany; 5Skull Base Research Center, Loghman Hakim Hospital, Shahid Beheshti University of Medical Sciences, Tehran 19835-35511, Iran

**Keywords:** lncRNA, miRNA, circRNA, pituitary gland cancer, meningioma

## Abstract

**Simple Summary:**

Non-coding RNAs have been recently attained attention because of their contribution in the pathogenesis of brain tumors. These transcripts have been shown to be dysregulated in pituitary gland tumors as well as meningiomas. In these two types of brain tumors, dysregulation of non-coding RNAs has been associated with some clinical features and response to therapeutic options. Different types of non-coding RNAs have been shown to interact with each other to promote progression of brain tumors. Further research is needed to find the possible application of non-coding RNAs as biomarkers for pituitary gland tumors as well as meningiomas, particularly in patients’ follow-up.

**Abstract:**

Long non-coding RNAs (lncRNAs), microRNAs (miRNAs), and circular RNAs (circRNAs) are non-coding transcripts which are involved in the pathogenesis of pituitary gland tumors. LncRNAs that participate in the pathogenesis of pituitary gland tumors mainly serve as sponges for miRNAs. CLRN1-AS1/miR-217, XIST/miR-424-5p, H19/miR-93a, LINC00473/miR-502-3p, SNHG7/miR-449a, MEG8/miR-454-3p, MEG3/miR-23b-3p, MEG3/miR-376B-3P, SNHG6/miR-944, PCAT6/miR-139-3p, lncRNA-m433s1/miR-433, TUG1/miR-187-3p, SNHG1/miR-187-3p, SNHG1/miR-302, SNHG1/miR-372, SNHG1/miR-373, and SNHG1/miR-520 are identified lncRNA/miRNA pairs that are involved in this process. Hsa_circ_0001368 and circOMA1 are two examples of circRNAs that contribute to the pathogenesis of pituitary gland tumors. Meanwhile, SNHG1, LINC00702, LINC00460, and MEG3 have been found to partake in the pathogenesis of meningioma. In the current review, we describe the role of non-coding RNAs in two types of brain tumors, i.e., pituitary tumors and meningioma.

## 1. Introduction

Non-coding RNAs comprise heterogeneous types of transcripts in terms of functions, size, evolutionary conservation, and expression level. The integrated application of large-scale sequencing methods and bioinformatics tools has facilitated the annotation of non-coding RNAs; thus, they are not considered as either junk portions of the genomes or byproducts of massive transcription. Soon after the completion of the Human Genome Project, several non-coding RNAs were detected in mammals [1]. With the advent of high-throughput sequencing strategies, the expression profile of non-coding RNAs has been more precisely identified [2]. The ENCODE project has stated that approximately 80% of the human genome is transcribed into non-coding RNAs [3]. These transcripts take part in manifold biological processes, controlling physiological and developmental events. Most notably, they have been recognized as tumor suppressors and oncogenes in numerous types of cancers [4]. Three classes of non-coding RNAs, i.e., long non-coding RNAs (lncRNAs), microRNAs (miRNAs), and, more recently, circular RNAs (circRNAs), have attained much attention in this field. LncRNAs are transcripts with more than 200 nucleotides and several shared features with mRNAs, yet lack detectable open reading frames [5]. They regulate the expression of genes by serving as signals, decoys, scaffolds, guide RNAs, or enhancer RNAs [5]. On the other hand, miRNAs are short-sized RNAs that mainly regulate the expression of genes at the post-transcriptional level through binding with 3′ UTR of transcripts and induction of gene silencing or mRNA degradation [6]. CircRNAs are produced mostly through the back-splicing of exons in precursor mRNAs. These covalently closed RNA transcripts function as miRNA sponges, serve as scaffolds for proteins, or modulate transcriptional and splicing events. Occasionally, circRNAs are used as templates for the synthesis of polypeptides. Abnormal expression of circRNAs has been reported in diverse malignancies. In addition to sustaining cell growth and proliferation, they promote tumor invasiveness and facilitate bypass of cell senescence and death [7]. In the current review, we describe the role of these classes of non-coding RNAs in two types of brain tumors, i.e., pituitary tumors and meningioma.

## 2. Pituitary Gland Tumors

Pituitary adenomas comprise a heterogeneous group of tumors historically classified based on their size into micro- and macroadenomas considering a threshold of 1 cm. More recently, immunohistochemical analyses and electron microscopy have been used to classify these tumors further [8]. These tumors can also be categorized as functional or nonfunctional, based on their hormonal activity. They might be identified either as an incidental finding in radiology or being associated with symptoms, particularly visual disturbance, based on their size, growth rate, or hormone secretion [9]. The overall prevalence of pituitary adenomas is estimated to be around 16% [10]. Recent studies have shown dysregulation of lncRNAs, miRNAs, and circRNAs in pituitary gland tumors. For instance, RPSAP52, an antisense lncRNA from the *HMGA2* locus, has been found to be overexpressed in both gonadotroph- and prolactin-producing adenomas of this gland, where its expression has been correlated with the expression of HMGA2. Contrariwise, the expression of RPSAP52 has been variable among somatotroph adenomas. RPSAP52 has been shown to act as a molecular sponge for miR-15a, miR-15b, and miR-16 to increase the expression of HMGA2. In addition, RPSAP52 could enhance the expression of HMGA1. RPSAP52 has been shown to enhance cell growth through increasing G1/S transition [11]. XIST is another upregulated lncRNA in pituitary tumors whose expression is increased parallel with upregulation of bFGF and downregulation of miR-424-5p. Functionally, XIST acts as a sponge for miR-424-5p to increase bFGF levels. Silencing of XIST or bFGF or overexpression of miR-424-5p has been shown to inhibit proliferation, migration, and invasiveness of pituitary neuroendocrine tumor cells [12].

### 2.1. LncRNAs

Several lncRNAs have been found to act as suppressors of tumorigenesis processes in pituitary adenomas. Among these, lncRNAs is clarin 1 antisense RNA 1 (CLRN1-AS1), which is underexpressed in prolactinomas. This lncRNA has a role in the inactivation of Wnt/β-catenin signaling. Being mainly located in the cytoplasm, CLRN1-AS1 has been shown to suppress cell proliferation, promote apoptosis, and inhibit autophagy. In addition, CLRN1-AS1 sponges miR-217 to increase the expression of the dickkopf WNT signaling pathway inhibitor 1 (DKK1). The FOXP1 transcription factor could suppress the expression of CLRN1-AS1 [13]. Additionally, H19 has been found to be commonly downregulated in primary pituitary adenoma samples. Downregulation of H19 has been correlated with tumor progression. Overexpression of H19 could suppress the proliferation of pituitary adenoma cells in vitro and their growth in vivo. Functionally, H19 could regulate tumorigenesis via suppressing the activity of mTORC1 but not mTORC2. In fact, H19 blocks mTORC1-mediated phosphorylation of 4E-BP1 without changing the activity of S6K1. H19 also interacts with 4E-BP1 and inhibits 4E-BP1 binding to Raptor. Notably, H19 could suppress pituitary tumors more effectively than cabergoline [14]. Moreover, H19 has been found to have a synergistic effect with dopamine agonists in prolactinoma. From a mechanistical point of view, H19 increases the expression of ATG7 through sequestering miR-93a [15]. Table 1 shows the list of lncRNAs participating in the pathogenesis of pituitary gland tumors. Figure 1 represents the role of several ncRNAs in regulating the MAPK/ERK, PI3K/AKT, Wnt/β-Catenin, and BMP cascades in pituitary gland tumors and meningiomas.

**Table 1 cancers-13-05987-t001:** LncRNAs and pituitary gland tumors (ANTs: adjacent non-tumor samples).

lncRNA	Accession Number/Location	Expression Pattern	Clinical Samples/Animal Model	Assessed Cell Lines	Targets/Regulators	Signaling Pathways	Description	Reference
RPSAP52	ENSG00000241749/12q14.3	↑	12 gonadotroph tumors and 3 normal samples	HT-29, HCT-116, BCPAP, GH3, ATt-20	HMGA2↑, HMGA1↑, miR-15a↓, miR-15b↓, miR-16↓	-	Microarray analysis and RT-PCR confirmed RPSAP52 upregulation, which enhances cell cycle progression and cell growth.	[11]
CLRN1-AS1	ENSG00000239265/3q25.1	↓	42 pairs of pituitary prolactinoma and ANTs, male BALB/C athymic nude mice	293T	miR-217↑, DKK1↓/FOXP1↑	Wnt/β-catenin	This lncRNA has the potential to limit cell proliferation and autophagy but enhance apoptosis rate. Additionally, FOXP1 diminishes CLRN1-AS1 transcription. CLRN1-AS1 downregulation was assessed by qRT-PCR.	[13]
XIST	ENSG00000229807/Xq13.2	↑	86 pituitary neuroendocrine tumors and 23 normal tissues	-	miR-424-5p↓, bFGF↑	-	Xist expression was evaluated by RT-qPCR. XIST downregulation reduces migration, invasion, cell cycle progression, proliferation capacity, and increased apoptosis rate.	[12]
H19	ENSG00000130600/11p15.5	↓	9 tumor and 9 normal tissues, athymic female nude mice	GH3, HEK293T	4E-BP1	-	H19 upregulation, evaluated by qRT-PCR reduces tumor progression and cell proliferation by inhibiting the mTORC1 normal function that mediates 4E-BP1 phosphorylation.	[14]
		↓	2 resistant and 3 sensitive prolactinoma tissues, female BALB/c athymic nude mice	GH3	miR-93a↓, ATG7↑	-	By sequestering miR-93a, H19 increases ATG7 expression, influencing resistance to dopamine agonists. H19 expression was evaluated by qRT-PCR.	[15]
LINC00473	ENSG00000112541/6q27	↑	Invasive and non-invasive pituitary adenoma, each with 20 cases, athymic female nude mice	AtT-20, GT1-1	miR-502-3p↓, KMT5A↑, cyclin D1↑, CDK2↑	-	LINC00473 increases cell cycle progression, proliferation, and tumor growth. RNA-sequencing and qRT-PCR were used to assess LINC00473 expression.	[16]
SNHG7	ENSG00000233016/9q34.3	↑	30 pituitary tumor and ANTs, nude mice	GH1, RC-4B/C, GH3, MMQ	miR-449a↓, Ki67↑	-	SNHG7 upregulation, evaluated by RT-PCR, results in enhanced cell migration, invasion, tumor growth, and reduced apoptosis.	[17]
MEG8	ENSG00000225746/14q32.2-q32.31	↑	20 bone-invasive and 20 non-invasive pituitry adenoma, male BALB/c nude mice	293T, RAW264.7	miR-454-3p↓, TNF-α↑	-	RT-PCR confirmed MEG8 upregulation, which leads to TNF-α increase. The increased TNF-α enhances osteoclast activity which results in increased bone destruction.	[18]
MEG3	ENSG00000214548/14q32.2	↓	34 tumors and ANTs	GH3, MMQ	miR-23b-3p↑, FOXO4↓	-	RT-qPCR indicated MEG3 reduction in tumors. MEG3 diminishes cell proliferation, invasion, migration, EMT, and upregulates apoptosis rate.	[19]
		↓	30 tumors and 12 normal pituitary tissues, nude mice	PDFS	miR-376B-3P↓, HMGA2↑	-	MEG3 downregulation was analyzed by qRT-PCR. Upregulation of MEG3 and MIR-376B-3P suppresses tumorigenesis and enhances apoptosis.	[20]
SNHG6	ENSG00000245910/8q13.1; 8q13	↑	Invasive and non-invasive pituitary tissues, each containing 30 cases.	HP75	miR-944↓, RAB11A↑	-	SNHG6 upregulation, measured by qRT-PCR, improves the proliferation, invasion, migration, viability, and EMT rate of tumor cells.	[21]
PCAT6	ENSG00000228288/1q32.1	↑	Tumors and ANTs from 20 invasive and 20 non-invasive cases, nude mice	RC-4B/C, GH3	miR-139-3p↓, BRD4↑	-	Located in the cytoplasm, PCAT6 increases cell proliferation, viability, invasion, cell cycle progression, migration but decreases apoptosis rate. RT-qPCR was used to evaluate PCAT6 expression.	[22]
lncRNA-m433s1	-/6q32	↑	Male and female SD rats	-	miR-433↓, Fshβ↑	-	RT-qPCR confirmed lncRNA-m433s1 upregulation. As an intergenic lncRNA, this non-coding RNA is located in the cytoplasm and upregulates follicle-stimulating hormone.	[23]
UCA1	ENSG00000214049/19p13.12	↑	30 pituitary tumors and 30 normal tissues	GH3, MMQ	prolactin↑, HK2↑, LDHA↑	-	qRT-PCR evaluation showed UCA1 overexpression. UCA1 enhances glycolysis, prolactin secretion, and cell growth.	[24]
C5orf66-AS1	ENSG00000249082/5q31.1	↓	11 patients and 4 normal cases	GT1-1	SCGB3A1↓	-	In this experiment RNA-sequencing, microarray analysis and qRT-PCR methods were used. C5orf66-AS1 upregulation leads to a marked reduction in cell viability and invasion.	[25]
PVT1	ENSG00000249859/8q24.21	↑	86 pituitary adenoma and ANTs	GH3, HP75, HNPG	β-Catenin↑, c-Myc↑, Cyclin D1↑	Wnt/β-catenin	qRT-PCR analysis indicated PVT1 upregulation. Cell proliferation, migration, and EMT are all improved by PVT1.	[26]
CCAT2	ENSG00000280997/8q24.21	↑	74 adenoma and corresponding normal tissues	HP75	PTTG1↑, SOX2↑, DLK1↑, MMP2↑, MMP13↑/E2F1	-	CCAT2 boosts cell proliferation, invasion, and migration but impedes apoptosis. Its expression was determined by RT-PCR.	[27]
TUG1	ENSG00000253352/22q12.2	↑	55 pituitary adenoma cases and 11 normal participants, BALB/c nude mice	HP75, GH3	miR-187-3p↓, p65↑, IκB-α↑, TESC↑	NF-κB	RT-qPCR assessed TUG1 expression. While TUG1 enhances the cell proliferation rate, invasion, migration, and EMT, the apoptosis rate is diminished.	[28]
AFAP1-AS1	ENSG00000272620/4p16.1	↑	60 pituitary adenoma and ANTs	GH3, MMQ	PTEN↓, PI3K↑, p-AKT↑	PTEN-PI3K-AKT	qRT-PCR showed a significant upregulation in AFAP1-AS1. Cell cycle progression and proliferation rates positively correlate with AFAP1-AS1 expression; in contrast to its negative correlation with the apoptosis rate.	[29]
IFNG-AS1	ENSG00000255733/12q15	↑	20 pituitary adenoma and ANTs	HP75	ESRP2↓	-	IFNG-AS1 overexpression, evaluated by qRT-PCR, significantly increases cell proliferation, invasion, migration, and lowers the apoptosis rate.	[30]
SNHG1	ENSG00000255717/11q12.3	↑	48 invasive and 10 non-invasive pituitary tumor tissues, nude mice	GH1, RC-4B/C	miR-302↓, miR-372↓, miR-373↓, miR-520↓, TGFBR2↑, RAB11A↑	Wnt/β-catenin, TGF-β	Although SNHG1 overexpression—appraised by RT-PCR—decreases the apoptosis rate, proliferation, cell cycle progression, invasion, migration, and EMT rates are all improved.	[31]

**Figure 1 cancers-13-05987-f001:**
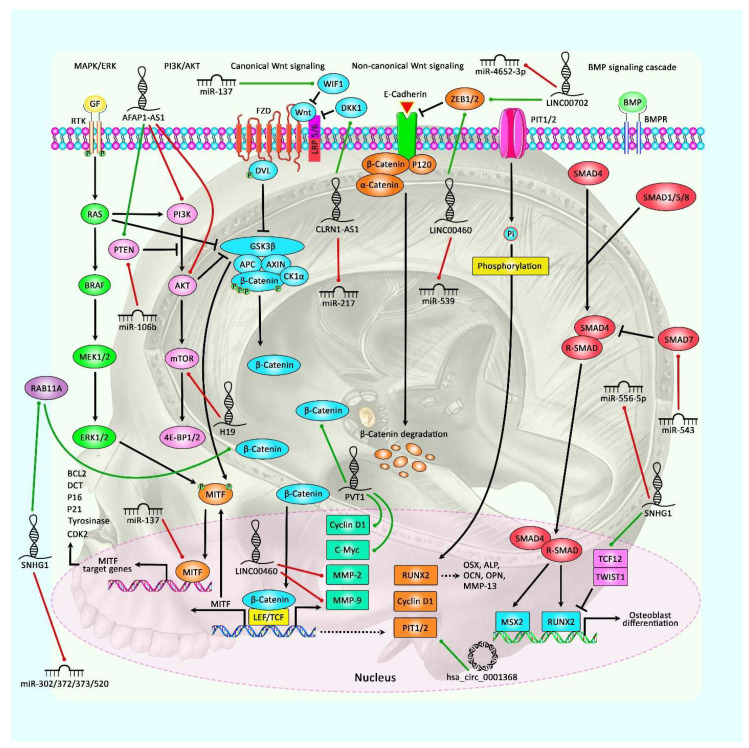
A schematic representation of the role of several non-coding RNAs in regulating the MAPK/ERK, PI3K/AKT, Wnt/β-Catenin, and BMP signaling pathways in pituitary gland tumors and meningiomas. The figure represents the potential crosstalk between various signaling cascades modulated via several ncRNAs in triggering the development of tumor cells. WNT-signaling is a crucial part of the crosstalk between key oncogenic cascades involved in pituitary gland tumors. Elements of the WNT cascades both could be regulated through diverse pathways, including MAPK/ERK, PI3K/AKT, and BMP, as well as transcriptional regulators containing p53 and MITF [32]. In addition, an accumulation of β-catenin in the cytoplasm could, in turn, lead to its translocation to the nucleus, where it could create a complex with TCF/LEF to trigger the transcription of RUNX2, Cyclin D1, and PIT1/2. Additionally, RUNX2 could modulate the transcription of various targets, including OCN, OSX, OPN, MMP13, and ALP [33]. Furthermore, BMP receptors could phosphorylate receptor-SMADs upon ligand binding. TCF12 and TWIST1 are basic helix-loop-helix transcription factors that could play an effective role in heterodimerization and suppressing transcription downstream of the BMP cascade [34]. According to the current report, lncRNA AFAP1-AS1 could enhance growth and inhibit apoptosis in pituitary adenomas through promoting PTEN expression and suppressing the expression levels of PI3K and AKT in tumor cells [29]. Moreover, another research has denoted that LINC00460 could elevate meningioma progression and metastasis through promoting the expression levels of MMP-2, MMP-9, and ZEB1 by sponging miR-539 and thereby acting as an oncogenic RNA in the meningioma malignancy [35]. Moreover, based on recent study, lncRNA SNHG1 via targeting miR-556-5p could elevate TCF12 expression, thereby promoting tumorigenesis of meningioma through the Wnt signaling cascade. In fact, TCF12 expression was positively modulated via SNHG1, and TCF12 could, in turn, enhance transcription of SNHG1 through binding with the promoter region of SNHG1 [36]. Green arrows indicate the upregulation of target genes modulated via ncRNAs (miRNAs and lncRNAs); red arrows depict inhibition regulated by them. All the information regarding the role of these ncRNAs in modulating pituitary gland tumors can be seen in Table 1, Table 2, Table 3, Table 4 and Table 5.

### 2.2. miRNAs

Dysregulation of miRNAs is also implicated in the pathogenesis of pituitary gland tumors. While almost all related miRNAs have been assessed in a single study, the expression of miR-34a and miR-145 has been evaluated in independent studies. Notably, the results of these studies are contradictory. Yang et al. have assessed the expression of miR-34a in rat pituitary tumor cells versus normal pituitary cells. The expression of this miRNA is lower in tumor cells compared with that in normal ones. Moreover, upregulation of miR-34a could suppress the proliferation of tumor cells and promote cell apoptosis via the regulation of expression of SOX7 [37]. On the other hand, another experiment in *AIP*mut+ cells as a genetic model of hereditary somatotropinoma has shown upregulation of miR-34a and miR-145 in *AIP*mut+ compared with *AIP*mut− somatotropinomas. Ectopic expression of *AIP*mut (p.R271W) in *Aip*^−/−^ mouse embryonic fibroblasts led to enhancement of miR-34a and miR-145 levels, demonstrating a pivotal correlation between *AIP*mut and miRNAs signature. Upregulation of miR-34a enhances proliferation, colony-forming ability, and migration of rat pituitary cells and inhibits their apoptosis. miR-145 could also affect the proliferation and apoptosis of these cells to a lesser extent. Overexpression of miR-34a enhances intracellular levels of the mitogenic factor cAMP and reduces octreotide-mediated growth hormone inhibition and antiproliferative impacts. miR-34a has been found to target the *Gnai2* gene, a gene that encodes a G protein subunit suppressing cAMP synthesis. Taken together, miR-34a has been identified as a downstream target of mutant *AIP* that confers a cellular phenotype reflecting the aggressive manifestations of *AIP*mut+ acromegaly [38].

Contrary to this report, the expression of miR-145-5p has been shown to be lower in bromocriptine-resistant prolactinoma clinical specimens and cell lines compared with that in sensitive samples and cells. TPT1 has also been identified as the direct target of miR-145-5p. Forced overexpression of miR-145-5p has increased the sensitivity of prolactinoma cells to bromocriptine through enhancing the expression of TPT1. Thus, miR-145-5p has been identified as a critical modulator of drug resistance in prolactinoma [39].

The expression of miR-378 has also been found to be decreased in pituitary adenoma tissues. This miRNA has been shown to downregulate the expression of RNF31. RNF31 silencing has remarkably inhibited the proliferation and migration of GH3 pituitary adenoma cells [40]. On the other hand, the expression of miR-543 has been found to be upregulated in pituitary adenoma tissues parallel with downregulation of Smad7. Smad7 has been verified as a target gene of miR-543. Overexpression of miR-543 has enhanced the proliferation, migration, and invasiveness of HP75 cells. This miRNA also reduces the apoptosis of these cells and decreases the expression of cleaved caspases 3 and 8 [41]. Table 2 shows the role of miRNAs in pituitary gland tumors. Figure 2 illustrates the role of various ncRNAs in pituitary gland tumors through regulating the TGF-β/SMAD signaling pathway.

**Table 2 cancers-13-05987-t002:** miRNAs and pituitary gland tumors (ANTs: adjacent non-tumor samples).

miRNA	Accession Number/Location	Pattern of Expression	Clinical Samples/Animal Model	Assessed Cell Lines	Targets/Regulators	Signaling Pathways	Description	Reference
miR-34a	ENSG00000284357/1p36.22	↓	female Rattus norvegicus	GH4C1	SOX7↑	-	miR-34a upregulation decreases cell proliferation and increases apoptosis. miR-34a downregulation was evaluated by qRT-PCR.	[37]
		↑	42 cases: 32 somatotropinomas and 10 prolactinomas, AIP knockout mice	GH3, HEK293, GH4C1	Gαi2↓, cAMP↑	-	Somatotropinomas with AIP mutations lead to enhanced miR-34a expression, upregulated intracellular cAMP concentration, and reduced Gαi2, resulting in somatostatin resistance. Additionally, miR-34a limits the apoptosis rate. To assess miRNA expression, microarray analysis and qRT-PCR were used.	[38]
miR-338-3p	ENSG00000283604/17q25.3	↑	10 microadenoma within sella turcica and 13 invading cavernous sinus macroadenoma cases	GH3	Pttg1↑, GH↑, prolactin↑	-	miR-338-3p improves cells’ invasiveness, migration, and proliferation rate. miRNA and qRT-PCR were used.	[42]
miR-378	ENSG00000199047/5q32	↓	25 tumors and ANTs	GH3	RNF31↑	-	RT-qPCR confirmed miR-378 downregulation. miR-378 reduces proliferation and migration rates.	[40]
miR-543	ENSG00000212040/14q32.31	↑	71 invasive and 66 non-invasive tumor tissues	HP75	Smad7↓	Wnt/β-catenin	miR-543 overexpression, appraised by RT-qPCR, resulted in improved cell proliferation, invasion, and migration but reduced apoptosis.	[41]
miR-134	ENSG00000207993/14q32.31	↓	29 patients affected by nonfunctioning pituitary neuroendocrine tumor	αT3-1	VEGFA↑/SDF-1α↑	-	Diminished levels of miR-134 were analyzed by qRT-PCR. SDF-1α decreases miR-134 and improves VEGFA to expand cell proliferation, viability, and cell cycle progression capacity.	[43]
miR-193a-3p	ENSG00000207614/17q11.2	↓	82 patients with pituitary adenoma: 42 nonfunctional, 32 prolactinomas, 5 growth hormone-secreting, and 2 follicle-stimulating hormone	-	-	-	qRT-PCR confirm miR-193a-3p downregulation. Its expression has a negative correlation with tumor size and recurrence rate.	[44]
miR-448	ENSG00000199001/Xq23	↓	Pituitary adenoma and ANTs	MMQ, HP75	BCL2↑	-	miR-448 downregulation was analyzed by qRT-PCR. Its overexpression restricts cell proliferation and migration and increases apoptosis.	[45]
miR-219a-2-3p	ENSG00000284185/9q34.11	↓	-	AtT-20, GT1.1, MPC	MDM2↑, p53↓	-	RT-qPCR was used to assess miR-219a-2-3p expression. miR-219a-2-3p improves apoptosis and inhibits cell proliferation.	[46]
miR-1299	ENSG00000275377/9p11.2	↑	12 drug-resistant and 6 sensitive patients	MMQ	FOXO1↓, prolactin↑	-	miR-1299 is upregulated in drug-resistant cases, which further inhibits FOXO1 expression. MicroRNA sequencing analysis and qRT-PCR were used in this experiment.	[47]
miR-410-3p	ENSG00000199092/14q32.31	↑	75 pituitary adenoma tissues: 34 gonadotroph, 30 somatotroph, 5 corticotroph, 3 plurihormonal, and 3 null cell tumors	RC-4B/C, AtT-20, GH3	cyclin B1↑, p14↓, Wee1↓	MAPK, PTEN/AKT, STAT3	Invasive tumors have a higher miR-410-3p expression level, which leads to a higher proliferation rate, invasion, and cell cycle progression rates in RC-4B/C and AtT-20 cells. GH3 cells showed a precisely opposite result. qRT-PCR was used to assess miR-410-3p expression.	[48]
miR-137	ENSG00000284202/1p21.3	↓	15 invasive and 15 non-invasive prolactinoma tissues, female F344 rats	MMQ, GH3	MITF↑, WIF-1↓	Wnt/β-catenin	miR-137 lowers cell proliferation, invasion, and β-catenin nuclear translocation rates. Tissue microarray and qRT-PCR were used in this project.	[49]
miR-205-5p	ENSG00000284485/1q32.2	↓	-	GH3, MMQ, HEK293T	CBX1↑	-	qRT-PCR confirmed miR-205-5p downregulation. miR-205-5p significantly reduces cell proliferation and migrations rates.	[50]
miR-93-5p	ENSG00000207757/7q22.1	↑	8 fibrous and 33 nonfibrous prolactinoma tissues	MMQ, HS27	Smad7↓, TGF-β1↑	TGF-β1/Smad3	Small RNA sequencing and qRT-PCR methods were used in this experiment. miR-93-5p induces fibrosis in prolactinoma cases through regulating the TGF-β1/Smad3 pathway.	[51]
miR-370	ENSG00000199005/14q32.31	↓	24 nonfunctional pituitary adenoma tissues	-	HMGA2↑/CXCL12↑	-	miR-370 expression negatively correlates with higher tumor grades and Ki-67-positive cells. It also restricts cell proliferation rate and boosts cell apoptosis rate. miR-370 downregulation was indicated by RT-PCR.	[52]
miR-145-5p	ENSG00000276365/5q32	↓	11 normal pituitary tissues, 24 bromocriptine-sensitive and 8 resistant samples, female nude mice	MMQ	TPT1↑	-	As confirmed by qRT-PCR, miR-145-5p is decreased in bromocriptine-sensitive tissues and highly reduced in bromocriptine-resistant tissues. On top of that, miR-145-5p upregulation reduces cell viability.	[39]
miR-16	ENSG00000208006/13q14.2	↓	36 patients and 8 healthy controls	HP75	p27↓, Bax↓, VEGFR2↓	NF-κB	RT-qPCR confirmed miR-16 downregulation. Cell proliferation and apoptosis rates are, respectively, positively and negatively correlated with miR-16 overexpression. In addition, this microRNA inhibits angiogenesis.	[53]
miR-124	ENSG00000284321/8p23.1	↓	68 invasive pituitary tissues: 7 growth hormone-secreting and 61 non-invasive pituitary adenomas	GH3	PTTG1IP↑/Cav-1, EGR1, KLF5	-	Caveolin-1 inhibits EGR1 translocation into the nucleus. Therefore, KLF5 does not interact with EGR1. Without the inhibitory effect of EGR1, KLF5 increases these miRNAs expression, which ultimately reduces PTTG1IP, FSCN1, and EZR levels—prohibiting cell invasion and migration. Microarray assay and qRT-PCR were used to assess miRNAs expression.	[54]
miR-145	ENSG00000276365/5q32	↓			FSCN1↑/Cav-1, EGR1, KLF5			
miR-183	ENSG00000207691/7q32.2	↓			EZR↑/Cav-1, EGR1, KLF5			
miR-148-3p	ENSG00000199085/7p15.2	↓	10 invasive and 10 non-invasive pituitary adenoma tissues	GH3, MMQ	ALCAM↑	-	The qRT- PCR method evaluated these miRNAs’ expression. Proliferation and invasion rates are reduced by these miRNAs, but apoptosis is enhanced.	[55]
miR-152	ENSG00000207947/17q21.32	↓						
miR-524-5p	ENSG00000283289/19q13.42	↓	20 adenoma and 8 normal tissues, BALB/c female nude mice	PDFS, HEK293FT	PBF↑	-	Proliferation, invasion, clonogenicity, tumor growth, and migration are all inversely correlated with miR-524-5p expression. qRT- PCR was used to analyze miR-524-5p expression.	[56]
miR-153	ENSG00000207647/2q35	↓	-	MMQ	Skp↑	-	miR-153 activates caspase-3 to increase apoptosis and decrease the proliferation rate.	[57]
miR-106b	ENSG00000208036/7q22.1	↑	32 invasive and 18 non-invasive adenoma tissues, 10 healthy control cases	HP75	PTEN↓	PI3K/AKT	Cell cycle progression, invasion, migration, and proliferation are markedly improved by miR-106b upregulation, which itself was analyzed by qRT-PCR.	[58]
miR-26a	ENSG00000199075/3p22.2	↑	12 normal, 31 invasive, and 39 non-invasive pituitary tissues	-	PLAG↓	-	Through downregulating PLAG, miR-26a enhances tumor invasiveness. miR-26a expression was evaluated by qRT-PCR.	[59]
miR-133	ENSG00000283927/18q11.2	↓	6 pituitary tumors and ANTs	HP75	FOXC1↑	-	miR-133 downregulation was assessed by RT-PCR. Cell migration, EMT, and invasion are inversely related to miR-133 expression.	[60]

### 2.3. circRNAs

Du et al. have assessed circRNAs signature in growth hormone-secreting pituitary adenoma using a circRNA microarray. They have reported upregulation of more than 1900 circRNAs and downregulation of about 1600 circRNAs in this type of adenoma compared with that in normal control. Ten most overexpressed circRNAs have been shown to be mainly enriched in the mTOR and the Wnt signaling pathway. Upregulation of hsa_circ_0001368 has also been confirmed by qRT-PCR. This circRNA has been found to be specifically overexpressed in this type of adenoma in correlation with the invasive properties and serum levels of growth hormone. Hsa_circ_0001368 silencing suppresses proliferation, invasion, and secretion of growth hormone from primary cultured cells. Additionally, levels of hsa_circ_0001368 have been positively correlated with the pituitary-specific transcription factor Pit-1 [62]. CircOMA1 is an upregulated circRNA in nonfunctioning pituitary adenomas which sponges miR-145-5p, a miRNA that inhibits the growth of this type of tumor through targeting TPT1 [63]. Table 3 shows the role of circRNAs in pituitary adenomas.

**Table 3 cancers-13-05987-t003:** circRNAs and pituitary gland tumors.

circRNA	Pattern of Expression	Clinical Samples/Animal Model	Assessed Cell Lines	Targets/Regulators	Signaling Pathways	Description	Reference
hsa_circ_0001368	↑	Growth hormone-secreting pituitary adenoma: 19, nonfunctioning pituitary adenoma: 20, prolactin-secreting adenoma: 18, ACTH-secreting adenoma: 12	-	Pit-1↑	mTOR, Wnt	Proliferation, invasion, and growth hormone-secretion levels are positively related to this circRNA expression. circRNA microarray, RNA-seq, and qRT-PCR were used in this project.	[62]
circOMA1	↑	50 nonfunctioning adenomas and 15 normal tissues, BALB/c nude mice	PDFS, HEK293T	miR-145-5p↓, TPT1↑, Mcl-1↑, Bcl-xL↑, Bax↓	-	The qRT-PCR method was used to appraise circOMA1 expression. Tumor invasion and cell proliferation are enhanced by circOMA1, whereas the apoptosis rate is decreased.	[63]

### 2.4. Prognostic/Diagnostic Value of Non-Coding RNAs in Pituitary Gland Tumors

The prognostic/diagnostic value of non-coding RNAs has been assessed in pituitary gland tumors (Table 4). Downregulation of SNHG7 [17], CCAT2 [27], and IFNG-AS1 [30] lncRNAs has been shown to increase the survival of patients with pituitary adenomas. miR-193a-3p upregulation results in a lower relapse-free survival rate [44]. On the other hand, upregulation of miR-137 expression is related to a higher recurrence-free survival rate [49].

**Table 4 cancers-13-05987-t004:** Prognostic/diagnostic value of non-coding RNAs in pituitary gland tumors (OS: overall survival, ANTs: adjacent non-tumor samples).

Non-Coding RNA	Accession Number/Location	Clinical Cases	AUC	Kaplan–Meier Analysis	Univariate/Multivariate Cox Regression	Reference
SNHG7	ENSG00000233016/9q34.3	30 pituitary tumors with high and low expression	-	A lower SNHG7 expression results in a higher OS rate.	-	[17]
CCAT2	ENSG00000280997/8q24.21	74 adenomas and corresponding normal tissues	-	Higher CCAT2 expression leads to a lower OS rate.	-	[27]
IFNG-AS1	ENSG00000255733/12q15	20 tumors and ANTs	-	Higher IFNG-AS1 expression is an indicator of a lower survival rate.	-	[30]
miR-193a-3p	ENSG00000207614/17q11.2	High: 29 Low: 53	-	miR-193a-3p upregulation leads to a lower relapse-free survival rate.	-	[44]
miR-137	ENSG00000284202/1p21.3	High: 16 Low: 14	-	Upregulated miR-137 expression is related to a higher recurrence-free survival rate.	-	[49]
miR-16	ENSG00000208006/13q14.2	36 patients were divided into high- and low-expression groups	-	miR-16 overexpression marks longer OS and disease-free survival rates.	-	[40]
miR-26a	ENSG00000199075/3p22.2	12 normal, 31 invasive and 39 non-invasive pituitary tissues	0.818	Downregulated miR-26a represents a shorter survival rate.	Tumor invasiveness, miR-26a, and PLAG1 expression could be used as survival risk factors.	[59]

## 3. Meningioma

Meningiomas are another group of brain tumors that are mostly encapsulated lesions. These benign lesions are typically associated with few types of genetic aberrations, yet their intracranial location can result in serious and possibly fatal consequences [64]. The expression of a number of lncRNAs and miRNAs has been dysregulated in meningioma.

### 3.1. LncRNAs

SNHG1 is an upregulated lncRNA in meningioma cell lines. SNHG1 silencing blocks cell growth and induces their apoptosis. SNHG1 could function as a sponge for miR-556-5p and enhance the expression of TCF12. In fact, the SNHG1/miR-556-5p/TCF12 axis could promote the proliferation of meningioma cells and suppress their apoptosis by enhancing the activity of Wnt signaling. Moreover, TCF12 has been shown to increase the expression of SNHG1 via binding with its promoter [36]. LINC00702 is another upregulated lncRNA in meningioma which regulates proliferation and migration of these cells via the miR-4652-3p/ZEB1 axis [65]. In addition, LINC00460 has been found to increase cell invasion and proliferation and decrease apoptosis rate through sponging miR-539 [35]. On the other hand, MEG3 is a possible tumor suppressor lncRNA in meningioma, which modulates invasive properties of these cells through the miR-29c/AKAP12 axis [66]. Table 5 shows the role of lncRNAs in meningiomas.

**Table 5 cancers-13-05987-t005:** LncRNAs and meningioma (ANTs: adjacent non-tumor samples).

lncRNA	Accession Number/Location	Pattern of Expression	Clinical Samples/Animal Model	Assessed Cell Lines	Targets/Regulators	Signaling Pathways	Description	Reference
SNHG1	ENSG00000255717/11q12.3	↑	-	CH157-MN, HBL-52, BEN-MEN-1, IOMM-Lee	miR-556-5p↓, TCF12↑/TCF12	Wnt/β-catenin	SNHG1 overexpression, indicated by qRT-PCR, has a marked impact on elevating cell proliferation and growth rates and inhibiting apoptosis.	[36]
LINC00702	ENSG00000233117/10p15.1	↑	88 malignant meningioma and ANTs	OMM-Lee, KT21, CH157-MN, HBL-52, Ben-Men-1	miR-4652-3p↓, ZEB1↑	Wnt/β-catenin	This lncRNA overexpression, as formerly showed by qRT-PCR, has a significant correlation with a lower OS rate. Cell migration and proliferation are positively associated with its expression.	[65]
LINC00460	ENSG00000233532/13q33.2	↑	33 meningioma and 10 normal meninges tissues	IOMM-Lee, CH157-MN, Ben-Men-1	miR-539↓, MMP-2↑, MMP-9↑, ZEB1↑	-	LINC00460 escalates cell invasion and proliferation and lowers the apoptosis rate. qRT-PCR was used to evaluate this lncRNA’s expression.	[35]
MEG3	ENSG00000214548/14q32.2	↓	5 healthy meninges and 32 meningioma tissues	(IOMM-Lee, CH157-MN	miR-29c↑, AKAP12↓	-	qRT-PCR showed MEG3 downregulation in tumor tissues. Cell cycle progression, proliferation, migration, and invasion are negatively correlated with MEG3 expression.	[66]

### 3.2. miRNAs

Negroni et al. have reported that the expression of the miR-497~195 cluster in meningioma is reduced with increasing tumor grade. Notably, Cyclin D1 upregulation has been correlated with a decrease in the levels of the miR-497~195 cluster. In fact, the effect of GATA binding protein 4 in enhancing the viability of meningioma cells is exerted through the regulation of expression of this miRNA cluster, which finally results in enhancement of Cyclin D1 level. Finally, serum exosome levels of miR-497 are lower in patients with high-grade meningioma compared to those with benign lesions [67].

Expression of miRNAs and mRNAs in benign and malignant meningiomas has also been assessed by RNA sequencing and miRNA microarray. This study has led to identifying upregulation of fatty acid synthase (FASN) in malignant lesions compared with benign ones. This gene has been shown to be targeted by miR-195. Overexpression of miR-195 could inhibit proliferation, migration, and invasiveness of malignant meningioma cells. Taken together, miR-195 has been verified as a tumor-suppressive miRNA in malignant meningioma by targeting FASN. NUP210, SPIRE2, SLC7A1, and DMTN are also among competing endogenous RNAs that modulate the expression of FASN through sponging miR-195 [68].

In another study, Katar et al. have shown significant enhancement in miR-21 levels with increasing grade of meningiomas, whereas there was a remarkable decrease in miR-107 levels with the increasing grade. Expressions of miR-137 and miR-29b have not been different in different histopathologic grades [69]. Table 6 shows the role of miRNAs in meningioma.

Finally, a number of studies have shown associations between genetic polymorphisms of lncRNAs or miRNAs and risk of meningioma (Table 7). For instance, the rs619586 A > G polymorphism of MALAT1 has been shown to affect expression levels of MALAT1 and COL5A1, resulting in lower invasiveness of meningioma [73]. Expression of MALAT1 has been shown to be reduced in a stepwise manner with enhancing levels of miR-145 in tumor/serum specimens having AA, AG, and GG genotypes of rs619586, respectively. In addition, levels of COL5A1 have been reduced in a similar stepwise manner in relation to the rs619586 genotypes. Thus, the rs619586A > G of the MALAT1 gene can decrease the expression of this lncRNA, influencing the impact of miR-145 on COL5A1. Consistently, meningioma cells harboring the G genotype of the rs619586 had higher levels of COL5A1 [73]. Moreover, certain haplotype blocks of miR-146a, miR-149, miR-196a2 and miR-499 have been shown to be associated with risk of meningioma [74].

## 4. Discussion

The contribution of lncRNAs, miRNAs, and circRNAs has been assessed in pituitary adenomas. lncRNAs that participate in the pathogenesis of pituitary gland tumors mainly serve as sponges for miRNAs. CLRN1-AS1/miR-217, XIST/miR-424-5p, H19/miR-93a, LINC00473/miR-502-3p, SNHG7/miR-449a, MEG8/miR-454-3p, MEG3/miR-23b-3p, MEG3/miR-376B-3P, SNHG6/miR-944, PCAT6/miR-139-3p, lncRNA-m433s1/miR-433, TUG1/miR-187-3p, SNHG1/miR-187-3p, SNHG1/miR-302, SNHG1/miR-372, SNHG1/miR-373, and SNHG1/miR-520 are identified lncRNA/miRNA pairs that are involved in this process.

The contribution of lncRNAs in meningioma is less studied. However, similar to pituitary adenomas, lncRNAs mainly serve as sponges for miRNAs. SNHG1/miR-556-5p, LINC00702/miR-4652-3p, LINC00460/miR-539, and MEG3/miR-29c are identified lncRNA/miRNA pairs in this type of lesion. Functional effects of miRNAs sponging are diverse, ranging from the enhancement of invasiveness of tumors to induction of epithelial-mesenchymal transition in these cells.

Exosomal levels of miRNAs in serum samples of patients have been associated with the presence of brain malignancies in some cases. However, the diagnostic and prognostic applications of these exosomes have not been assessed comprehensively. In other types of cancer, exosomal levels of non-coding RNAs represent an applicable source of biomarkers for prediction of course of cancer as well as response to therapies [75].

Expression of non-coding RNAs might also affect response to therapeutic regimens [76]. For instance, H19 increases ATG7 expression and influences resistance to dopamine agonists [15].

Theoretically, the expression of non-coding RNAs can be used for molecular classification of meningiomas. However, this approach has not been implemented yet. A previous study has used the DNA methylation signature for this purpose. This approach has captured clinically more homogenous groups. Moreover, it has been proved to be superior to WHO classification in prediction of tumor recurrence and prognosis [77].

Genetic variants within non-coding regions have the potential to affect the function of these transcripts or the regulatory impacts of other transcripts on non-coding ones. A number of these variants have been demonstrated to be associated with the risk of meningioma, yet their impacts on the risk of pituitary tumors have not been revealed.

Since aberrant expressions of lncRNAs/miRNAs can affect the response of tumor cells to therapeutic modalities, it is possible that targeted therapies for modulation of expression of lncRNAs/miRNAs not only reduce the invasiveness of these tumors but also increase their response to conventional therapies.

## 5. Conclusions

Cumulatively, non-coding RNAs represent an emerging class of transcripts with putative effects on the pathogenesis of pituitary adenomas as well as meningiomas. Future studies are needed to find possible specific markers for each of these tumors to help in the identification of tumors in a less invasive manner.

## Figures and Tables

**Figure 2 cancers-13-05987-f002:**
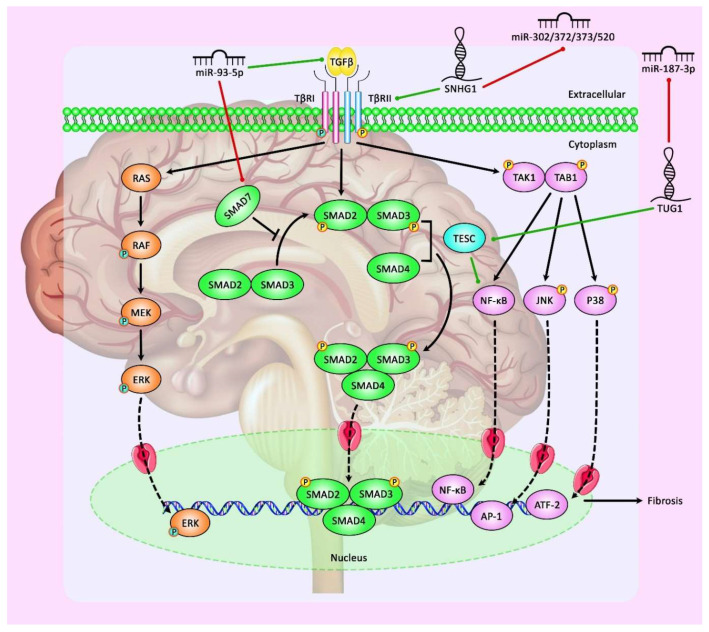
A schematic diagram of the role of various ncRNAs in modulating the TGF-β/SMAD signaling pathway in pituitary gland tumors. According to this cascade, it could be triggered through the binding of active TGF-β with TβRII and forming the TβRI-TβRII heteromeric complex, resulting in phosphorylation of Smad2/3, oligomerization with Smad4, and consequent nuclear translocation to modulate the transcription of ECM genes. Furthermore, Smad7 could play a remarkable role as a negative modulator of the TGF-β cascade. In addition, TGF-β has a significant part in triggering the activation of downstream signaling pathways containing MAPK, modulated by the Ras-Raf-MEK-ERK cascade, and TAK1, regulated by the TAB1 pathway. This could also lead to mediating the activation of MKK4-JNK and MKK3-p38 cascades and upregulation of AP-1 and ATF-2, respectively, and the overexpression of NF- κB to modulate profibrotic responses [61]. Previous studies have authenticated that several ncRNAs could have a significant part in regulating the TGF-β/SMAD cascade in pituitary gland tumors. As an illustration, recent literature has detected that overexpression of lnc-SNHG1 could considerably elevate the expression level of TβRII through activating TβRII/SMAD3 in invasive pituitary tumor cells via sponging miR-302/372/373/520 [31]. Furthermore, other research has indicated that upregulation of miR-93-5p could downregulate the expression level of Smad7, thereby activating the TGF-β1/Smad3 signaling-mediated fibrosis of prolactinoma cells [51]. Green arrows indicate the upregulation of target genes modulated via ncRNAs (miRNAs and lncRNAs); red arrows depict inhibition regulated by them.

**Table 6 cancers-13-05987-t006:** miRNAs and meningioma.

miRNA	Accession Number/Location	Pattern of Expression	Clinical Samples/Animal Model	Assessed Cell Lines	Targets/Regulators	Description	Reference
miR-497~195	ENSG00000267532/17p13.1	↓	80 meningioma and 25 primary meningioma cases	KT21-MG1-Luc5D, Ben-Men-1	Cyclin D1↑/GATA-4↑	GATA-4 upregulation restricts miR-497~195 cluster expression and increases cell viability. RT-PCR was used to assess miR-497 expression.	[67]
miR-195	ENSG00000284112/17p13.1	↓	3 paired malignant and benign meningioma cases	IOMM-Lee	FASN↑/NUP210, SPIRE2, SLC7A1, DMTN	Migration, invasion, and proliferation rates of tumor cells are axiomatically elevated by miR-195 downregulation. RNA-sequencing, miRNA microarray, and qRT-PCR methods were used in this experiment.	[68]
miR-21	ENSG00000284190/17q23.1	↑	50 patients affected with meningioma	-	-	miR-21 and miR-107 have positive and negative correlations with tumor grade, respectively. Their expression was evaluated by miRNA detection kit.	[69]
miR-107	ENSG00000198997/10q23.31	↓					
miR-34a-3p	ENSG00000284357/1p36.22	↓	35 meningioma cases	Ben-Men-1, HEK293T	SMAD4↑, FRAT1↑, BCL2↑	miR-34a-3p downregulation was assessed by RT-PCR. While the proliferation rate is diminished after miR-34a-3p overexpression, the apoptosis rate is improved.	[70]
miR-29c-3p	ENSG00000284214/1q32.2	↑	58 meningioma tumor tissues	MEN-117, MEN-141	PTX3↓	Microarray analysis and RT-PCR were used in this experiment. Cell viability is improved by miR-29c-3p upregulation. In contrast, apoptosis is lowered by miR-29c-3p.	[71]
let-7d	ENSG00000199133/9q22.32	↓	17 meningioma samples	IOMM-Lee, CH-157MN	AEG-1↑	qRT-PCR evaluated let-7d expression. Proliferation, invasion, and viability are effectively inhibited by let-7d, whereas the apoptosis rate is elevated.	[72]

**Table 7 cancers-13-05987-t007:** Polymorphisms of non-coding RNAs in meningioma.

lncRNA	Accession Number/Location	Clinical Samples	Assessed Cell Lines	Polymorphism	Description	References
MALAT1	ENSG00000251562/11q13.1	427 invasive and 402 non-invasive meningioma cases	KNS-89, SNB-19	A > G (rs619586)	AA genotype increases invasive meningioma risk.	[73]
miR-146a	ENSG00000283733/5q33.3	69 meningioma and 183 healthy controls	-	C > G (rs2910164)	These three haplotypes significantly increase the chances of meningioma: (1) miR-146a-miR-149-miR-196a2 -miR-499: G-T-C-G (2) miR-146a-miR-196a2 -miR-499: G-C-G (3) miR-149-miR-196a2 -miR-499: C-C-G	[74]
miR-149	ENSG00000207611/2q37.3			T > C (rs4846049)		
miR-196a2	ENSG00000207924/12q13.13			T > C (rs11614913)		
miR-499	ENSG00000207635/20q11.22			A > G (rs3746444)		

## Data Availability

The analyzed data sets generated during the study are available from the corresponding author upon reasonable request.

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
