# Peer review of "The Emerging Role of Non-Coding RNAs in Pituitary Gland Tumors and Meningioma"

_cancers, 2021, doi:10.3390/cancers13235987_

Round 1
Reviewer 1 Report
The authors propose a quite interesting review about non-coding RNAs in two types of tumor: PitNet and meningiomas. It is quite questionable to have regrouped these two type of lesions in the same manuscript, because these pathology are very different.
I would suggest significant improvements in the review methodology, structuration of the manuscript and focus on potential clinical perspectives.
Suggestions :
- It would be better to present this work as a systematic review, with mat. meth. section and refer to PRISMA checklist
- Describe, after the introduction molecular techniques that were used in the different studies
- Develop the potential applications of these ncRNA as prognostic or predictive markers and as therapeutical targets (e.g. refer to preclinical or clinical studies in other types of tumors)
- Structure the different sections with subsections (i.e. miR, lnc, circ, clincal perspectives)
- PitNET and meningioma are not cancers, excepting high grade lesions: change cancer or carcinogenic or cancerous with tumor / tumorigenesis or progression, throughout the text and tables
- Page 2, line 64: 16% of prevalence seems to be very high, please verify this data
- Page 2, line 94: mechanistical
Author Response
- Thank you for your comment. This review is a narrative review. Since the number of ncRNAs is high and most of them have been assessed in only one original article, it was not possible to have a meta-analysis.
- We added the molecular techniques in the Tables.
- We discussed prognostic or predictive values of these ncRNAs.
- We organized the text into subsections.
- We corrected this point.
- The authors found an overall estimated prevalence of pituitary adenomas of 16.7% (14.4% in autopsy studies and 22.5% in radiologic studies).
- We corrected the mentioned point.
Reviewer 2 Report
I acknowledge the authors to have performed an exhaustive review on this interesting topic.
However, several questions should be discussed:
- what about the methods of sequencing? Have the authors observed heterogeneities of methods in the reported studies? Are the reported results influenced by these differences? The discussion should be developed regarding this point.
- The clinical (location of tumours, methods used in the studies, follow-up time, retrospective or prospective series) and histological (WHO grade in meningioma patients) data should be reported in the manuscript, namely in the tables,
- What do the authors think about other approaches like DNA methylation, e.g. in meningiomas? Have these approaches (DNA methylation and non coding RNAs) been assessed in terms of clinical impact (risk of recurrence, for instance)?
- What about the risk of resistance to treatments related to the RNA profiles (for instance, resistance to ionizing radiation)? Are there specific data about this question?
- Were there correlations between the radiological aspect of tumours in MRI and the RNA profiles?
I propose the tables and figures to be corrected:
- in a same table, reporting of in vitro and in vivo (mice, humans) data is not relevant; maybe separate tables would be more appropriate,
- I don't think that mentioning bibliographic references in the titles of the figures is necessary,
- Why are meningiomas reported in the title of the figure 1 which should deal with the pathways described in pituitary adenomas?
- The presentation of the tables should be corrected (some tables on several pages).
Author Response
- Most of studies have assessed expression of non-coding RNAs using qRT-PCR.
- We added data in the tables whenever it was reported in the main articles.
- We could not find any data about the impact of DNA methylation and non coding RNAs on clinical outcome.
- We added some notes about the impact of lncRNAs on response to dopamine agonists. There was no data about radioresistance.
- There was no data about correlations between the radiological aspect of tumours in MRI and the RNA profiles.
- Figure 1 shows important pathways in both pituitary gland tumors and meningiomas. We corrected the title.
- We corrected the presentation of tables. Further corrections should be done during typesetting.
Round 2
Reviewer 2 Report
I acknowledge the authors to have considered some of the previous propositions.
Nevertheless, I suggest the presentation of the tables could be improved (tables in a separate section and not inserted in the main text). I also recommend to cite and discuss about the article published by Sahm et al. (Lancet Oncol 2017) which deals with a potential DNA methylation-based classification of meningiomas. Would such a sort of classification feasible using non coding RNAs?
Page 17, lines 285-287: please cite one or several bibliographic reference(s) to illustrate this sentence.
Author Response
- We put tables in separate sections.
- We cited this article and discussed it.
- We added two references for this section.